# Development and Validation of a Genotypic Assay to Quantify CXCR4- and CCR5-Tropic Human Immunodeficiency Virus Type-1 (HIV-1) Populations and a Comparison to Trofile^®^

**DOI:** 10.3390/v16040510

**Published:** 2024-03-27

**Authors:** Daisy Ko, Sherry McLaughlin, Wenjie Deng, James I. Mullins, Joan Dragavon, Socorro Harb, Robert W. Coombs, Lisa M. Frenkel

**Affiliations:** 1Center for Global Infectious Disease Research, Seattle Children’s Research Institute, Seattle, WA 98109, USA; daisy.ko@seattlechildrens.org (D.K.);; 2Department of Microbiology, University of Washington, Seattle, WA 98195, USA; dengw@uw.edu (W.D.); jmullins@uw.edu (J.I.M.); 3Department of Medicine, University of Washington, Seattle, WA 98104, USA; 4Department of Global Health, University of Washington, Seattle, WA 98105, USA; 5Department of Laboratory Medicine and Pathology, University of Washington, Seattle, WA 98195, USA; jdragavon2021@gmail.com (J.D.); socoharb@gmail.com (S.H.); bcoombs@uw.edu (R.W.C.); 6Department of Pediatrics, University of Washington, Seattle, WA 98195, USA

**Keywords:** X4 virus, CXCR4 co-receptor, R5 virus, CCR5 co-receptor, maraviroc

## Abstract

HIV-1 typically infects cells via the CD4 receptor and CCR5 or CXCR4 co-receptors. Maraviroc is a CCR5-specific viral entry inhibitor; knowledge of viral co-receptor specificity is important prior to usage. We developed and validated an economical V3-*env* Illumina-based assay to detect and quantify the frequency of viruses utilizing each co-receptor. Plasma from 54 HIV+ participants (subtype B) was tested. The viral template cDNA was generated from plasma RNA with unique molecular identifiers (UMIs). The sequences were aligned and collapsed by the UMIs with a custom bioinformatics pipeline. Co-receptor usage, determined by codon analysis and online phenotype predictors PSSM and Geno2pheno, were compared to existing Trofile^®^ data. The cost of V3-UMI was tallied. The sequences interpreted by Geno2pheno using the most conservative cut-off, a 2% false-positive-rate (FPR), predicted CXCR4 usage with the greatest sensitivity (76%) and specificity (100%); PSSM and codon analysis had similar sensitivity and lower specificity. Discordant Trofile^®^ and genotypic results were more common when participants had specimens from different dates analyzed by either assay. V3-UMI reagents cost USD$62/specimen. A batch of ≤20 specimens required 5 h of technical time across 1.5 days. V3-UMI predicts HIV tropism at a sensitivity and specificity similar to those of Trofile^®^, is relatively inexpensive, and could be performed by most central laboratories. The adoption of V3-UMI could expand HIV drug therapeutic options in lower-resource settings that currently do not have access to phenotypic HIV tropism testing.

## 1. Introduction

Human immunodeficiency virus type-1 (HIV) enters host cells after the viral envelope protein (env) engages the CD4 receptor and then typically one of two chemokine co-receptors: CC chemokine receptor 5 (CCR5) or CXC chemokine receptor 4 (CXCR4). Most people living with HIV have viruses that use CCR5 (R5) during the early and asymptomatic stages of infection, but the virus population often evolves to become dual- or mixed-tropic (DM) or use CXCR4 (X4) exclusively during or shortly prior to the onset of AIDS [1,2,3,4]. Maraviroc is a CCR5 co-receptor antagonist that can be administered orally to block R5-tropic viruses from using CCR5, but it does not block DM or X4 viruses [5,6,7,8].

To maximize the potential efficacy of maraviroc, pre-treatment screening for X4 viruses was explored using the Trofile^®^ assay and by next-generation sequencing for the detection of X4 viruses. It was found that a frequency of ≥2% within an individual’s HIV population was associated with virologic failure [9,10,11]. Pre-treatment screening is therefore recommended, with maraviroc reserved for individuals without detectable X4 variants [12]. Trofile^®^ creates patient-specific pseudoviruses containing a PCR-amplified region of their viral envelope, which is then cultured with co-receptor-expressing cells in the presence of maraviroc. Trofile^®^, only performed at Monogram Biosciences^®^ in Foster City, California, has a turnaround time of 28–35 days and is costly.

Multiple parallel sequencing technologies serve as alternatives to Trofile^®^, offering the potential for higher throughput, shorter turnaround times, and lower costs [9,13,14,15,16]. However, sequencing can introduce and magnify errors due to the following: (A) imperfect fidelity of the reverse transcriptase and DNA polymerases used, (B) sub-optimal cycling conditions that create incomplete, shortened templates that can act as primers in subsequent PCR cycles or PCR recombination, and (C) preferential amplification of certain templates [17]. Sequencing errors can also be introduced from the following: (A) misreading of fluorescence emissions between A and C, and G and T; and (B) reading frame errors when a cluster of templates becomes asynchronous with other clusters as the sequencing reaction cycles [17]. The incorporation of unique molecular identifiers (UMIs) allows the creation of a consensus sequence from reads with the same UMI [18]. This overcomes most PCR and sequencing errors and provides a count of the number of viral templates sequenced from an HIV population [18,19,20,21]. In this regard, UMI-based sequencing is superior to the Trofile^®^ approach, since the latter does not correct for errors during PCR or preferential amplification of certain templates.

We describe the development and cost of a high-throughput, genotype-based protocol using UMIs to quantify and identify the tropism of HIV, based on env V3 (third variable loop) sequences within plasma specimens (“V3-UMI”) and the validation of V3-UMI compared to Trofile^®^.

## 2. Materials and Methods

Between 2008 and 2017, remnant plasma specimens collected from 54 individuals living with HIV and tested by Trofile^®^ were utilized (as approved by the University of Washington Institutional Review Board). HIV subtypes and clinical viral loads (VL) were gathered from clinical records. Specimens were de-identified by a clinician.

### 2.1. RNA Extraction, cDNA Generation and HIV Env V3 Amplification

RNA was extracted from plasma using the QIAamp^®^ Viral RNA Mini Kit (Qiagen, LLC., Germantown, MD, USA) according to the manufacturer’s protocol with the following modifications: lysis of plasma with VL ≥ 10,000 copies/mL was scaled up 2× (280 µL plasma into 1.12 mL lysis buffer); when VL < 10,000 copies/mL, virions were concentrated prior to extraction by centrifugation of 1–1.25 mL of plasma for 30 min at ~77,000 rcf at 4 °C in a fixed-angle TLA-110 rotor (Beckman Coulter Life Sciences, Inc., Indianapolis, IN, USA). RNA was eluted in 50 µL of water and immediately converted into cDNA.

Reverse transcription reagents (SuperScript^®^ III Reverse Transcriptase, ThermoFisher Scientific Inc., Waltham, MA, USA) were scaled up to an 80 µL reaction to generate cDNA from the full 50 µL RNA eluate. RNA was incubated with 0.25 µM of cDNA primer and 0.5 mM of dNTP mix for 1 min at 65 °C, followed by a brief incubation on ice. The reaction volume was completed by the addition of the following: 1× SuperScript^®^ III FirstStrand Buffer, 1.25 mM DTT, and 20 U RNase inhibitor (Promega, Inc., Madison, WI, USA). In addition, 200 U of SuperScript^®^ III reverse transcriptase was added, followed by incubation at 50 °C for 30 min.

The cDNA primer included the following segments (Table 1): an HIV envC3-specific binding site (HXB2 positions 7377-7353), a 12-nucleotide semi-random UMI (with six positions restrictedly randomized to 2–3 bases to decrease mispriming with assay primers), and primer-binding sites for the first and second round PCR (Integrated DNA Technologies, Inc., Coralville, IA, USA). Primers targeting HIV were designed using an alignment of >500 subtype B sequences from the United States (Los Alamos National Laboratories HIV Database) (Table 2) and vetted for optimal ∆G profiles and the absence of predicted primer dimers and hairpin loops. The second round primer included MiSeq adapters (Table 1).

Following cDNA generation, ExoI (New England Biolabs, Inc., Ipswich, MA, USA) was added to the RNA:cDNA hybrid to degrade excess cDNA primers (30 min at 37 °C), followed by heat inactivation (15 min at 80 °C). Short fragments were removed by AMPureXP^®^ magnetic beads (Beckman Coulter, Inc.) using 1.2× of the reaction volume, following manufacturer instructions. cDNA was eluted in 20 µL of H_2_O and immediately transferred to first round PCR.

A quantitative PCR (qPCR) amplified and assessed the total number of HIV RNA templates reverse transcribed into cDNA, with the forward primer annealing to the relatively conserved env C2 region and the reverse primer to the “ill.1” part of cDNA reverse transcription primer (Table 1). Quantification standards used the env-containing plasmid p2-7 DNA [22]. The full volume of eluted cDNA (20 µL) was divided between four replicate 50 µL qPCR reactions using 0.75X SensiMix™ (Meridian Bioscience Inc., Cincinnati, OH, USA) and 0.3 µM primers (env6880F and ill.1; Table 1). qPCR conditions were as follows: 10 min at 95 °C, then 50 cycles of 10 s at 92 °C; 20 s at 60 °C; 20 s at 72 °C, ending with a standard melt curve. The yield was compared to clinical VL to assess efficiency of reverse transcription using our cDNA primer. 

Products from the first-round qPCR replicates were pooled to total ≤10,000 HIV templates. The mixture was diluted 1:20, and 1 µL was added to a 20 µL nested second-round PCR containing primers to incorporate MiSeq™ adapter sequences, 1× MyTaq™ buffer, 2.5 U MyTaq™ polymerase (Meridian Biosciences Inc., Cincinnati, Ohio, USA), and 0.2 µM primers (envC2F6 and ill.2 with MiSeq™ adapter sequences; Table 1). Thermal cycling conditions were the following: 5 min at 94 °C, then 20 cycles of 20 s at 94 °C; 20 s at 60 °C; 20 s at 72 °C, and a final extension of 7 min at 72 °C. PCR products were confirmed by gel electrophoresis, then purified with 0.8× AMPure^®^ XP magnetic beads (Beckman Coulter Inc.) for the sequencing of the 498 bp HIV V3 region of env.

PCR amplicons were indexed with the Nextera™ XT Index Kit (Illumina Inc., San Diego, CA, USA) using 2X KAPA HiFi™ Hotstart ReadyMix (Roche Diagnostics Ltd., Basel, Switzerland) and cleaned with AMPure^®^ XP beads using 0.8× of the reaction volume. Amplicon sizes were verified by gel electrophoresis. Each indexed PCR library was quantified using the Quant-iT™ PicoGreen™ dsDNA Assay Kit (ThermoFisher Inc., Waltham, MA, USA) and normalized to 5 nM. Libraries were pooled, quantified with the Quant-iT™ PicoGreen^®^ DNA Assay Kit (ThermoFisher Inc.), then loaded onto the MiSeq™ cartridge for sequencing with MiSeq™ Reagent Kit V3, 600 cycles (Illumina Inc.). To assess the efficiency of the second-round PCR and sequencing, the input template frequency determined by qPCR was compared to unique UMIs detected after all analyses.

### 2.2. MiSeq™ Data Analysis

Raw sequence reads were processed with a custom bioinformatics pipeline available on GitHub (https://github.com/MullinsLab/MiSeq_viral_tropism, accessed on multiple dates, with final date of 12 March 2022). This pipeline included the following steps: sequence quality filtering was performed with Sickle, primers and UMI were trimmed with Cutadapt, then paired reads were merged with PEAR [23,24,25]. Next, each UMI was extracted, and its associated env sequences were merged, with a read-number cut-off model applied that excluded offspring UMIs to determine the minimum number of sequences required to create a template consensus sequence [20]. Sequences were then aligned with MUSCLE v3.8.31, and a template consensus sequence was built with a majority variant (>50%) [26,27]. Lastly, V3 loop sequences between 84 and 126 nt were extracted and collapsed into unique variants. The pipeline output was a FASTA file containing all unique V3 sequence variants identified in a specimen, with each sequence name containing the total count of originating templates with that sequence, as determined by UMI analysis.

Cross-contamination was assessed by phylogenetic analysis of aligned sequences using Geneious 8.0.3 (https://www.geneious.com). Final trees were generated using the PhyML 3.3 and GTR substitution model options in DIVEIN (https://indra.mullins.microbiol.washington.edu/DIVEIN/, accessed on multiple dates, with final date of 12 March 2022), using each unique variant defined by the UMI [28,29]. Rarely, a few sequences from one specimen clustered with another, indicating index-hopping or cross-contamination; these were excluded from analyses [30].

### 2.3. HIV Tropism Determined by PSSM, Geno2pheno and Codon Analysis 

FASTA file outputs containing unique V3 sequences were queried with PSSM (https://indra.mullins.microbiol.washington.edu/webpssm/) and Geno2pheno 2.5 (https://coreceptor.geno2pheno.org/, accessed on multiple dates, with final date of 12 March 2022) [31,32]. The output of PSSM included a score and predicted tropism, whereas Geno2pheno predictions were characterized by various false positive rates (FPR). PSSM and Geno2pheno outputs were processed with custom perl scripts to calculate total template counts and tropism percentages (https://github.com/MullinsLab/MiSeq_viral_tropism, accessed on multiple dates, with final date of 12 March 2022). Additionally, amino acids at codons 11, 24, and 25 were examined, with basic (positively charged) amino acids at any of these positions used to define a virus as X4 (referred to below as the codon rule) [33]. The percentage of X4 sequences within a specimen was calculated from the number of estimated X4 viral templates divided by total viral templates determined from the unique UMI count. Sequences were analyzed by five cut-offs across a Geno2pheno FPR of 2 through 5.75. Sensitivity, specificity, positive, and negative predictive values of genotypic classifications of the specimens tested were compared to Trofile^®^ findings.

### 2.4. Cost of Assay

The reagent cost to sequence one specimen in a batch of ~20 specimens and the technician time required to perform the protocol are reported.

## 3. Results

### 3.1. Clinical Plasma HIV RNA and HIV Template Recovery

Remnant plasma from 54 individuals with Trofile^®^ results, all HIV subtype-B, was analyzed by V3-UMI. Trofile^®^ classified 32 (59%) as R5 and 22 (41%) as X4 or DM (Figure 1). Of the 54 specimens analyzed by V3-UMI, 33 (61%) were aliquots of the same plasma tested by Trofile^®^ (of which 25 had available clinical VL), and 21 (39%) were from a different date. The number of HIV templates submitted to reverse transcription, as estimated by clinical VL, correlated with the yield of cDNA, quantified by first-round qPCR (Pearson’s correlation coefficient = 0.210). If we exclude a possible sample mix-up (PID #49 discussed later), Pearson’s correlation coefficient becomes 0.704 (Figure 2A). The number of HIV templates sequenced from each specimen, quantified by UMI, correlated with the number of cDNA templates submitted for PCR/sequencing (Pearson’s correlation coefficient = 0.927) (Figure 2B). Two of the 54 specimens yielded insufficient amplicons or sequences and were excluded from subsequent analyses; these included PID #54, with a clinical VL of 76 c/mL, which yielded four cDNA templates identified by qPCR; and PID #51, with a clinical VL of 309 c/mL, which yielded two consensus sequences identified by UMI (Figure 1). The clinical VL from the 52 participants (including 20 from a different date than the specimens sent for Trofile^®^ and seven unavailable) were significantly greater when V3-UMI yielded sequences from ≥500 HIV templates vs. <500 templates (median 46,600 c/mL or 4.67 log10 (n = 37) vs. 17,685 c/mL or 4.25 log10 (n = 8), respectively; *p* = 0.003 by unequal variances t test. However, the available clinical VL were similar from specimens classified by Trofile^®^ as X4/DM vs. R5 (medians = 32,450 vs. 45,890 c/mL; log10 4.51 vs. 4.66 c/mL, n = 21 and 26, respectively).

### 3.2. Comparison of V3-UMI Sequences Interpreted by Genotypic Analysis to Trofile^®^

The tropism classifications for most specimens were similar between V3-UMI genotypic interpretation algorithms and Trofile^®^ (Figure 1), particularly for the 31 participants for whom both Trofile^®^ and V3-UMI were performed on aliquots from the same specimen date. The sensitivity, specificity, positive (PPV), and negative predictive values (NPV) of V3-UMI compared to Trofile^®^ are shown for all 52 specimens (Table 3) and for the subset of 31 specimens for which Trofile^®^ and V3-UMI were performed on plasma from the same date (Table 4). Across the three genotypic interpretation algorithms, Geno2pheno at a 2% FPR cut-off provided the closest agreement with Trofile^®^ across all 52 participants, with 90% (47/52) of specimens in agreement as either R5 or X4/DM. While the 2% FPR cut-off had the highest PPV (100%), it missed five of 21 (24%) specimens classified as X4/DM by Trofile^®^ (sensitivity of 76%); however, four of these assessed tropisms using plasma from a different date or misattributed specimen, and the fifth switched to X4 at Geno2pheno 3% FPR (PID #52) (Figure 1). A comparison of PSSM analyses to Trofile^®^ found 83% (43/52) of specimens in agreement as either R5 or X4, with a PPV of 77% and sensitivity 81% (four specimens classified as X4 by Trofile^®^ were classified as R5). Codon analysis yielded a PPV of 84% and a sensitivity of 76%. Across the algorithms, 33 of 52 (63.5%) specimens were fully concordant (19 R5, 14 X4) and 19 (36.5%) specimens were partially discordant for R5-X4 classification. Within the subset of 31 specimens with plasma from the same date assessed by Trofile^®^ and V3-UMI (n = 31), 20 (64.5%) were fully concordant (9 R5, 11 X4) and 11 (35.5%) were partially discordant (Table 4).

### 3.3. Analysis of Classifications Discordant by Trofile^®^ vs. Genotypic Algorithms

A review of the X4/DM discordant results between Trofile^®^ and Geno2pheno at 2% FPR (n = 5/52; PID#47, #48, #49, #50, and #52, as shown in Figure 1, found that three (#47, 48, 50) had specimens analyzed by V3-UMI at a later date than Trofile^®^ (2.5 weeks for #47, 4 weeks for #48, and 28 weeks for #50); #50 and #52 were identified as X4 by PSSM and codon analysis. Another, PID #49, had a clinical VL below the limit of detection (<LOD), yet UMI sequenced 14,113 templates and the maximum likelihood phylogram grouped all sequences in one large R5 clade (Figure 3A); both findings suggest a sample mix-up. Lastly, PID#52 was classified as X4 by Trofile^®^, based on the codon rule (4400/4401 sequences with Arginine at position 25) and PSSM (14% = X4), but only classified as X4 by Geno2pheno at a 3% FPR (0.1% = X4 at 2% FPR); the maximum likelihood phylogram for PID#52 had both X4 and R5 clades (Figure 3B). When plasmas from the same date were analyzed by Trofile^®^ and V3-UMI (n = 31), tropism classifications were concordant for X4 by Trofile^®^ and Geno2pheno at 2% FPR in 11 of 13 (84.6%) specimens; the two discordant specimens were PID#49, with the suspected sample mix-up, and PID#52, which switched to X4 at Geno2pheno at a 3% FPR. Reviewing participants (n = 19/52; 36.5%) with “partially discordant” tropism by Trofile^®^ and one or more of the genotypic analyses identified an additional discordance pattern: specimens were classified as R5 by Trofile^®^ but as X4 by PSSM, codon rule or by Geno2pheno at >2% FPR. These included PID#45, classified as R5 by Trofile^®^. Despite having 1448 V3-UMI sequences that Geno2pheno classified as R5 at a 2% FPR, it was classified as X4 by PSSM, codon rule, and Geno2pheno, at ≥ 3% FPR (X4 = 4.7% X4) (Figure 1); #46 was classified as R5 by Trofile^®^ and Geno2pheno at 2% FPR (1.8% of 541 V3-UMI sequences were X4) but switched to X4 at Geno2pheno 3% FPR (14.8% = X4); in addition, it was classified as X4 by PSSM (14.8% = X4) and the codon rule (84% = X4) (Figure 1). Phylograms of #45 and #46 had clades of both X4 and R5 sequences (Figure 3C).

### 3.4. Cost Analysis

The cost of reagents and supplies for V3-UMI is approximately $62USD/specimen. The processing of a batch of ~20 specimens takes ~1.5 h/specimen across 1.5 days. Sequence analyses with the pipeline were performed in <5 min/specimen, with additional time required to calculate the frequency of X4/DM templates in an individual’s HIV population. In contrast, Trofile^®^ performed at the Monogram Biosciences^®^ in Foster City, California has a turnaround time of 28–35 days, and quoted costs can be >$2500/specimen.

## 4. Discussion

We developed a genotypic assay (V3-UMI) that can quantify the frequency of X4 within an HIV plasma population, allowing the identification of those specimens with the clinically significant level of ≥2% X4 [9,14]. The testing of specimens by both V3-UMI and Trofile^®^ yielded comparable results, suggesting that V3-UMI could be used in place of this commercial phenotypic assay to predict HIV-1 susceptibility to the CCR5 co-receptor antagonist maraviroc. Multiple design features of V3-UMI result in the following: (1) high sensitivity and specificity compared to Trofile^®^; (2) feasibility to perform in the most central laboratories; and (3) a lower cost than Trofile^®^.

The HIV primers (cDNA and PCR) used in V3-UMI aim to minimize primer bias by use of relatively conserved regions of HIV subtype B envC2/C3 and the incorporation of the reverse qPCR primer binding sites into the cDNA primer. In addition, the assay reduces the chance of mispriming by omitting the UMI likely to bind to PCR primers. This is achieved by restricting randomized bases at six positions in the UMI with use of non-N degenerate bases. Generating consensus sequences using UMIs eliminates most errors from PCR and sequencing, providing a count of HIV templates sequenced [18]. This allows for an assessment of whether a sufficient number of viral templates were sequenced to confidently detect X4 variants at ≥2% of an individual’s HIV population. Sequence analysis to categorize as X4 or R5 can be conducted by several open-source pipelines. PSSM and codon analysis are simple to perform, as sequences are categorically classified as R5 or X4. Additionally, PSSM provides a score that indicates intermediate evolution from R5 to X4 tropism. The Geno2pheno algorithm requires the user to write code or manually upload sequences (currently, there is no automatic interface) and to choose a FPR cut-off to define viruses as R5 or X4, which then categorizes sequences as R5 or X4; however, changing the FPR allows detection of viral templates that appear to be evolving from R5 at a low FPR to X4 at a higher FPR. The preferred pipeline may vary by the HIV subtype of the specimens analyzed, as Geno2pheno was originally trained with HIV subtype B, and PSSM includes versions for subtypes B and C [31,33].

Geno2pheno, at a 2% FPR cut-off, had the highest specificity and positive predictive value across all 52 participants. In addition, it demonstrated even better concordance among a subset of 31 participants from whom we used aliquots from the same specimen for both Trofile^®^ and V3-UMI. This subset excluded discordant results, likely due to analyses of specimens from different dates (PID#47, #48, #50), as well as an apparent specimen mix-up (PID#49), which unfortunately can occur [34].

A weakness of both the Trofile^®^ and PCR-based assays for tropism is that the primers used to generate cDNA and amplify the virus via PCR may not anneal to certain HIV variants. This primer bias may have contributed to the discordance observed in only 1 (3%) of the 31 specimens tested by both Trofile^®^ and V3-UMI (PID#52; Figure 1). Trofile^®^ is not known to quantify R5 or X4 variants, so it is possible that Trofile^®^ assessed a greater number of total or X4 templates for PID#52, allowing detection of X4 below our ≥2% cut-off. Importantly, input cDNA from most specimens analyzed correlated with participants’ clinical VL, suggesting that V3-UMI primers generated and amplified cDNA from most viral templates regardless of tropism in most cases. Additionally, specimens with low-frequency X4 appear to have been occasionally missed by Trofile^®^ (specimens #46 and #45), potentially due to inadequate sampling of the patient’s virus population or primer bias.

A potential weakness with genotype-based assays is false-positive designation of sequences as X4. When using a Geno2pheno FPR cut-off of 2%, V3-UMI specimens classified as X4 were also X4 by Trofile^®^. However, higher Geno2pheno FPR thresholds resulted in discordance compared to Trofile^®^, as was the case with the use of PSSM and codon analysis, despite the presence of mutations associated with X4 tropism and genetic distances closer to sequences classified as X4 at the 2% cut-off. It is possible that these sequences were not included among those analyzed by Trofile^®^, or that HIV envelopes transitioning to X4 do not bind CXCR4 as strongly as fully transitioned envelopes to the indicator cells used in Trofile^®^, and thus were missed. The clinical outcomes of participants in our study are unknown; however, upon reanalysis of the maraviroc MOTIVATE trial specimens, a Geno2pheno 5% FPR was used to correlate that ≥2% X4 of an individual’s HIV population would result in virologic failure [14].

As with other multiple parallel sequencing protocols, cross-contamination of viral templates can occur during plasma extraction, cDNA generation, PCR, or can appear due to index hopping [17,30]. In addition to following good laboratory practices, such as unidirectional workspaces within the laboratory, filtered pipette tips, and clean benchwork practices, we used dual indexes to recognize and exclude index hopping, as well as phylogenetic analysis to identify cross-contamination.

## 5. Conclusions

Maraviroc, an antiretroviral that blocks HIV from utilizing the CCR5 co-receptor for cell entry, has proven utility and a good safety profile [35,36] but is underutilized, in large part due to the unavailability or cost of the Trofile^®^ assay commonly used to detect X4 variants prior to treatment. The V3-UMI genotypic assay is sensitive and specific for detection of X4 variants at low frequencies within an individual’s HIV population. V3-UMI could be performed in many central laboratories in mid- and low-resource settings. Studies to evaluate the use of V3-UMI for treatment regimens that include maraviroc combined with other drug classes may be warranted.

## Figures and Tables

**Figure 1 viruses-16-00510-f001:**
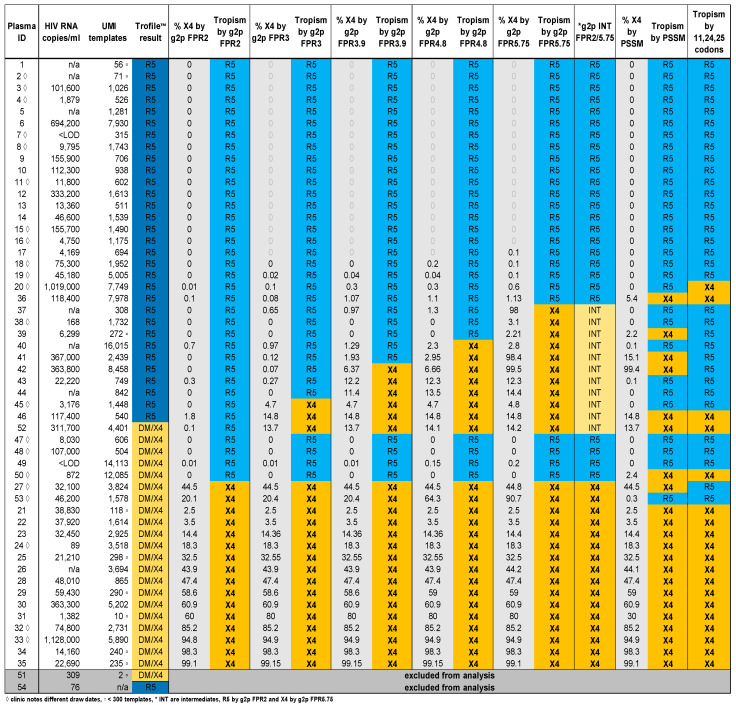
Participants’ plasma HIV clinical viral loads with Trofile^®^ and V3-UMI results (n = 54). Columns from left: participant ID, clinical plasma HIV RNA load, number of templates sequenced by V3-UMI, Trofile^®^ classification, X4 template frequency (%) calculated using a custom pipeline by five Geno2pheno levels in ascending FPR, PSSM, and analysis of codons 11/24/25. Blue shading indicates R5 and yellow indicates X4 by corresponding analytic program; ◊ indicates plasma from a different timepoint was used for clinical HIV RNA viral load testing; ▫ indicates <300 templates were recovered by UMI analysis. Gray shading of PIDs #51 and #54 indicates exclusion from assessing for X4 due to too few V3-UMI templates.

**Figure 2 viruses-16-00510-f002:**
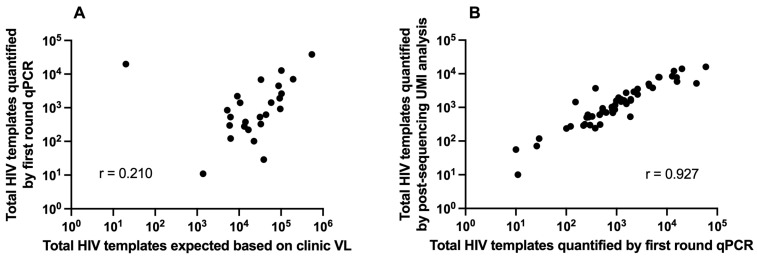
Efficiency of HIV template amplification. HIV RNA templates subjected to the assay, as estimated from the clinical plasma HIV RNA load (VL) (x-axis), were compared to the templates recovered following RNA extraction and reverse transcription, as measured by qPCR (y-axis) in (**A**). Clinical VL was available for 45 specimens; however, 20 of these were from a different date than the Trofile^®^ test specimen; the remaining 25 were plotted here. The outlier in (**A**) is PID #49, a possible sample mix-up described in the text. Pearson’s correlation coefficient between clinical HIV RNA VL and cDNA recovered by qPCR was 0.210 (0.704 excluding PID #49), indicating ~1 log10 fewer HIV cDNA templates compared to the input quantity, as determined by clinical plasma HIV RNA load. cDNA templates, as determined by qPCR (x-axis) were compared to the frequency of templates sequenced by V3-UMI (y-axis) for 52 specimens in (**B**); Pearson’s correlation coefficient was 0.927, indicating an approximately similar number of HIV templates sequenced compared to the input quantity of cDNA.

**Figure 3 viruses-16-00510-f003:**
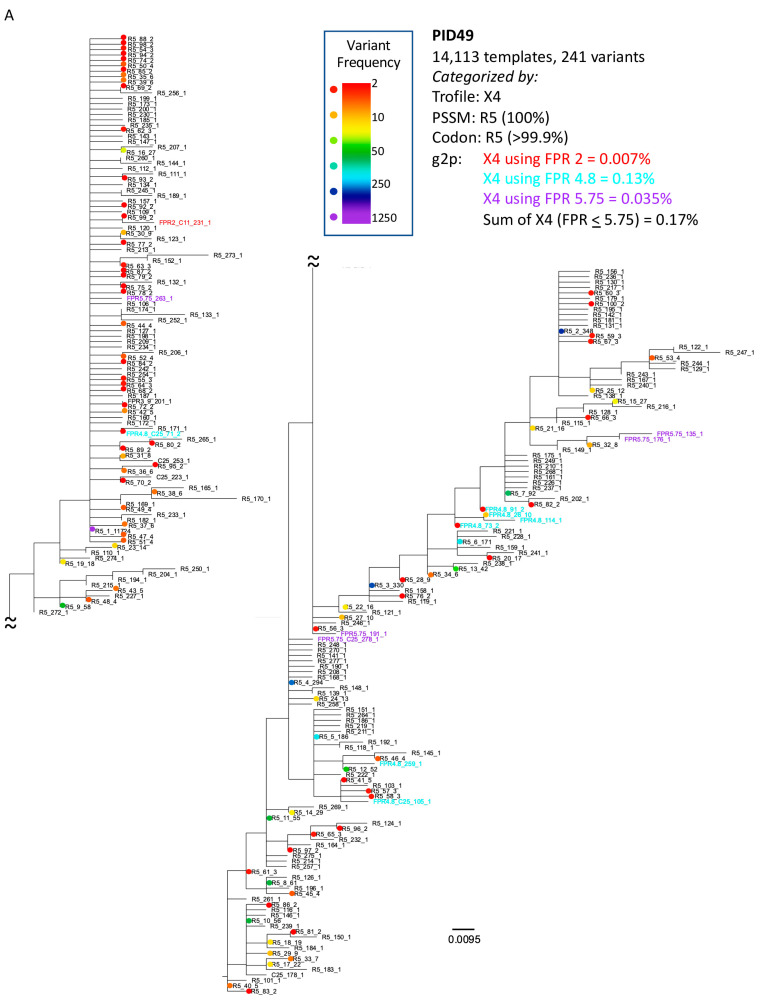
Maximum-likelihood phylogenetic trees of select specimens discordant by Trofile^®^ and X4-UMI. Trees are made from unique V3 genotypes aligned with MUSCLE using Geneious 8.0.3 and generated with DIVEIN PhyML v3.3 (https://indra.mullins.microbiol.washington.edu/DIVEIN/diver.html, accessed on multiple dates, with final date of 12 March 2022) [28]. Unique V3 genotypes were often comprised of multiple templates, defined by multiple unique molecular identifiers (UMI). Frequency of each variant as determined by UMI analysis is shown in colored circles according to the color gradient chart, and by the number at the far right of the sequence name. Tropism, as determined by geno2pheno FPR cut-off, is shown by the text color of each sequence name. Sequences predicted to be X4 by PSSM and codon analysis are indicated in the name of each sequence (“PSSM”, and “C#” where the # corresponds to the codon positions). The last two values in each sequence name, separated by underscores, correspond to the sequence identifier, followed by the quantity of templates associated with that unique V3 variant, as determined by UMI analysis. (**A**) Specimen #49 was categorized as X4 by Trofile^®^ and R5 by all genotypic analyses (given that plasma HIV RNA load, as determined by our qPCR of the specimen we sequenced, differed by >1 log_10_ from that reported to us, and the tropism calls by Trofile^®^ differed from all genotypic interpretation algorithms, we suspect a specimen mix-up occurred at some point prior to our specimen processing). (**B**) Specimen #52 was categorized as X4 by Trofile^®^, PSSM, codon analysis, and geno2pheno cut-offs ≥3% FPR, but was identified as R5 by geno2pheno at 2% FPR. (**C**) Specimens #45 and #46 were both categorized as R5 by Trofile^®^; geno2pheno classified both specimens as R5 at FPR 2% but transitioned to X4 at FPR 3%. Specimen #45 was classified as R5 and specimen #46 as X4 by PSSM and codon analysis, demonstrating the potential added value of the continuous scale used by Geno2pheno.

**Table 1 viruses-16-00510-t001:** Primer description, map, and sequences.

Primer	Purpose	Sequence
cDNA	cDNA synthesis	5′-CCCGCGTGGCCTCCTGAATTAT[**ill.1**]-CCGCTCCGTCCGACGACTCACTATA[ill.2]-SSMVSBYNNNNN[**UMI**]-CAGTAGAAAAATTCCCCTCCACAATT[**HIV7377-7353**] -3′
env6880F	1st round, forward	5′-CCCCGGCTGGTTTTGCGATTCTAAAGTGTA[**HIV6880-6909**] -3′
ill.1	1st round, reverse	5′-CCCGCGTGGCCTCCTGAATTAT -3′
envC2F6	2nd round, forward	5′-TCGTCGGCAGCGTCAGATGTGTATAAGAGACAG[**MiSeq adapter**]-GCACAGTACAATGTACACATGGAATTA[**HIV6952-6978**] -3′
ill.2	2nd round, reverse	5′-GTCTCGTGGGCTCGGAGATGTGTATAAGAGACAG[**MiSeq adapter**]-CCGCTCCGTCCGACGACTCACTATA[**ill.2**] -3′

**Table 2 viruses-16-00510-t002:** Percent identity of V3-UMI primers at each position of HIV-specific regions in cDNA (top) and first-round qPCR primers (bottom).

**cDNA oligo, 5′-3′ (HXB2 7353-7377)**														
99%	99%	90%	95%	99%	95%	99%	99%	90%	99%	99%	90%	99%		
5′-C	A	G	T	A	G	A	A	A	A	A	T	T-		
99%	90%	99%	99%	95%	95%	90%	95%	99%	99%	95%	95%	99%		
-C	C	C	C	T	C	C	A	C	A	A	T	T-3′		
**env6880, 5′-3′ (HXB2 6880-6909)**														
99%	99%	99%	90%	75%	99%	99%	99%	99%	99%	95%	99%	95%	99%	99%
5′-C	C	C	C	G	G	C	T	G	G	T	T	T	T	G-
99%	90%	90%	99%	95%	90%	99%	95%	95%	95%	70%	99%	99%	95%	99%
-C	G	A	T	T	C	T	A	A	A	G	T	G	T	A-3′

**Table 3 viruses-16-00510-t003:** Performance of genotypic models compared to Trofile® phenotypic assay.

Model. (n = 52)	Sensitivity (%)	False Positives	Specificity (%)	False Negatives	Positive Predictive Value (PPV)	Negative Predictive Value (NPV)
g2p FPR 2	76%	0	100%	5	100%	86%
g2p FPR 3	81%	2	93.5%	4	89.5%	88%
g2p FPR 3.9	81%	5	84%	4	77%	87%
g2p FPR 4.8	81%	7	77%	4	71%	86%
g2p FPR 5.75	81%	10	68%	4	63%	84%
PSSM	81%	5	84%	4	77%	87%
codon analysis	76%	3	90%	5	84%	85%

**Table 4 viruses-16-00510-t004:** Performance of genotypic models compared to Trofile^®^ phenotypic assay: subset of specimens with Trofile^®^ and genotypic analysis performed on the same date as shown in Table 3.

Model (n = 31)	Sensitivity (%)	False Positives	Specificity (%)	False Negatives	Positive Predictive Value (PPV)	Negative Predictive Value (NPV)
g2p FPR 2	85%	0	100%	2	100%	90%
g2p FPR 3	92%	1	94%	1	92%	94%
g2p FPR 3.9	92%	4	78%	1	75%	93%
g2p FPR 4.8	92%	6	67%	1	67%	92%
g2p FPR 5.75	92%	8	56%	1	60%	91%
PSSM	92%	5	72%	1	71%	93%
codon analysis	92%	2	89%	1	86%	94%

## Data Availability

All data generated for this study are presented in the figures and tables, except for sequence data, which are available at the NCBI Sequence Read Archive under BioProject number PRJNA1003530.

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
