# Peer review of "Development and Validation of a Genotypic Assay to Quantify CXCR4- and CCR5-Tropic Human Immunodeficiency Virus Type-1 (HIV-1) Populations and a Comparison to Trofile®"

_viruses, 2024, doi:10.3390/v16040510_

Round 1

Reviewer 1 Report

Comments and Suggestions for Authors

In this study, Ko et al. described a protocol of using an inexpensive UMI-based NGS to detect HIV X4 virus, and compared several X4 interpretation algorithms with phenotyping assay, the Trofile assay. The paper is well-written and the assay the author used was described in details. I really appreciate the authors did the comparison of UMI numbers and the template numbers. This type of comparison is critical to demonstrate the accuracy of the UMI clustering algorithms, but often lacking in most of the papers using UMI for NGS. Here are my comments and suggestions.

1. What is the senstitivity of the Trofile assay with low-level viremia? Trofile assay requires viral load greater than 1,000 cp/mL. It was surprising to see Trofile assay generated data for PID 51 and 54.

2. The authors used 2% intra-host frequency cut-off for calling X4 virus. It is unclear what cut-off Trofile uses, but Trofile claims that they can detect X4 variants as low as 0.3%. Please justify the use of 2% intra-host frequency for the sequencing assay.

3. Another way to think about calling out X4 virus at low frequency is to calculate the detection sensitivity given the number of UMI sequences.  Use #40 as an example, it had 16015 UMI sequences, it should have 95% confidence to detect a minority variant as low as 0.023% based on binomial distribution. (calculated using R function `binom.test(0,16015)`). It may overall call X4 virus for Maraviroc sensitivity but for other purposes, these low-frequency X4 viruses may not be ignored. For instance, if the patient ever goes for stem cell translation from a delta-32 homozygous donor, these low frequency X4 may rebound.

3. You authors may want to comment on the lack of API from Geno2pheno, which makes it very difficult to automate the interpretation of X4 virus using Geno2pheno. Using packages like `MechanicalSoup` is not ideal because single change of the Geno2pheno webpage may cause errors.

4. Can the authors identify the potential source of contamination of sample 49?

Author Response

see uploaded file

Reviewer 2 Report

Comments and Suggestions for Authors

The authors have developed a new assay to determine the proportion of HIV virions circulating in an individual that are either R5 (CCR5) or X4 (CXCR4)-tropic. Such a determination is useful when opting for the use of Maraviroc (a CCR5 inhibitor) as HIV therapy. The assay is based on sequence information, avoids some of the pitfalls of RT-PCR based determinations and is compared in the manuscript to the standard assay, which requires the preparation and testing of pseudotypes, but takes much longer.

The bulk of the manuscript is clear. Overall, the authors have demonstrated that there is good, though not perfect concordance between the standard Trofile assay and their analysis. They provide an extensive explanation of the instances of discordance, indicating that in some cases the samples were obtained on different dates. They propose that one sample is discordant because of a sample mix-up. Figure 3A is clearer in the new version, but still bit of a trial for a sample that the authors believe is incorrect, and perhaps should be eliminated and the explanation just included in the text.

Author Response

In this study, Ko et al. described a protocol of using an inexpensive UMI-based NGS to detect HIV X4 virus, and compared several X4 interpretation algorithms with phenotyping assay, the Trofile assay. The paper is well-written and the assay the author used was described in details. I really appreciate the authors did the comparison of UMI numbers and the template numbers. This type of comparison is critical to demonstrate the accuracy of the UMI clustering algorithms, but often lacking in most of the papers using UMI for NGS. Here are my comments and suggestions.

 Reviewer 1

  1. What is the sensitivity of the Trofile assay with low-level viremia? Trofile assay requires viral load greater than 1,000 cp/mL. It was surprising to see Trofile assay generated data for PID 51 and 54.

We agree that given the reported low levels of plasma HIV RNA for specimens from participants 51 and 54 that it is surprising that Trofile gave results. As the Reviewer states, Monogram Biosciences reports that the in vitro sensitivity of the Trofile assay is 0.3%. These data come from in vitro comparisons of the original assay to methods modified to amplify and submit a greater number of viral templates to their assay, presented in a poster (Antivir Ther. 2008;13(Suppl 3):A128). At the time the modified assay was called the “enhanced” assay. To our knowledge, these data were not published in a manuscript, and we do not know whether Monogram or others have examined the sensitivity of the Trofile assay for specimens with plasma HIV RNA load <1000c/mL.

  1. The authors used 2% intra-host frequency cut-off for calling X4 virus. It is unclear what cut-off Trofile uses, but Trofile claims that they can detect X4 variants as low as 0.3%. Please justify the use of 2% intra-host frequency for the sequencing assay.

We used the intra-host X4 frequency of 2% to define a “clinically significant” quantity of X4 HIV based on past reports; reference numbers 9-11. However, we agree with the Reviewer that data are meager to substantiate 2% as “the clinically significant” level of X4 that will cause maraviroc to fail to suppress HIV replication. Given that current ART regimens combine multiple antiretrovirals, and that except for parenteral ART adherence to therapy is difficult to evaluate, that it is unlikely that formal studies will confirm or refine the clinical cut-off for X4 variants (2% or other?).

  1. Another way to think about calling out X4 virus at low frequency is to calculate the detection sensitivity given the number of UMI sequences.  Use #40 as an example, it had 16015 UMI sequences, it should have 95% confidence to detect a minority variant as low as 0.023% based on binomial distribution. (calculated using R function `binom.test(0,16015)`). It may overall call X4 virus for Maraviroc sensitivity but for other purposes, these low-frequency X4 viruses may not be ignored. For instance, if the patient ever goes for stem cell translation from a delta-32 homozygous donor, these low frequency X4 may rebound.

We appreciate and agree with the Reviewer’s point that the 95% CI for detection of X4 variants may provide relevant calls. We debated on whether to report these values, but because the ≥2% cut-off for X4-tropic virus was clinically validated, we decided to classify our results by this cut-off. However, we think that the cutoff for clinically significant X4-tropic virus will likely vary depending on the clinical situation. We agree that it is unknown if X4 variants at ≤2% will replicate in persons given transplants from CCR5 delta-32 homozygous donors. Thus, we think the Reviewer points out a gap in our knowledge regarding the lower cut-off for categorizing X4 subpopulations as clinically significant.

  1. You authors may want to comment on the lack of API from Geno2pheno, which makes it very difficult to automate the interpretation of X4 virus using Geno2pheno. Using packages like `MechanicalSoup` is not ideal because single change of the Geno2pheno webpage may cause errors.

We agree with the Reviewer that use of Geno2Pheno is tedious. We propose revising the text of the manuscript, as follows, to mention that the user interface is inadequate.

                            Propose changing sentence beginning in line 315 to:

The Geno2pheno algorithm requires the user to write code or manually upload sequences (currently no automatic interface) and to choose a FPR cut-off to define viruses as R5 or X4, which then categorizes sequences as R5 or X4, and by changing the FPR allows detection of viral templates that appear to be evolving from R5 at low FPR to X4 at higher FPR.

  1. Can the authors identify the potential source of contamination of sample 49?

We did not identify the precise specimen mix-up that led to discrepent results for Sample #49. Our detection of significantly more HIV RNA in the specimen we sequenced (>1 log10) compared to the clinic-reported plasma HIV RNA (14,113 templates/mL vs. <LOD, respectively) together with the categorization of the specimen as X4 by the Trofile assay while all genotypic determinations were for R5 tropic virus suggested the possibility of a specimen mix-up. Phylogenetic analyses of sequences from all specimens processed in our lab for this and concomitantly conducted projects found no mix-up in our lab, but because the specimens were de-identified, we could not investigate further. Therefore, we think it was likely that specimens were mixed-up at some point prior to our specimen processing. We have added text to Figure 3 legend to better explain these findings.

Reviewer 2

The authors have developed a new assay to determine the proportion of HIV virions circulating in an individual that are either R5 (CCR5) or X4 (CXCR4)-tropic. Such a determination is useful when opting for the use of Maraviroc (a CCR5 inhibitor) as HIV therapy. The assay is based on sequence information, avoids some of the pitfalls of RT-PCR based determinations and is compared in the manuscript to the standard assay, which requires the preparation and testing of pseudotypes, but takes much longer.

The bulk of the manuscript is clear. Overall, the authors have demonstrated that there is good, though not perfect concordance between the standard Trofile assay and their analysis. They provide an extensive explanation of the instances of discordance (includes Participants 45, 46, 49, 52), indicating that in some cases the samples were obtained on different dates. They propose that one sample is discordant because of a sample mix-up (Participants 49). Figure 3A is clearer in the new version, but still bit of a trial for a sample that the authors believe is incorrect, and perhaps should be eliminated and the explanation just included in the text.

We appreciate the Reviewers comment that Figure 3a is difficult to navigate. We considered the Reviewer’s suggestion to remove the presentation of these data. However, to be transparent with readers, we have opted to leave data from Participant 49 in the manuscript. To improve the clarity by which we relay the data, we are submitting a new version of Figure 3a with improved resolution, and we revised the figure legend to provide additional details – proposed revision:

Figure 3. Maximum-likelihood phylogenetic trees of select specimens discordant by Trofile® and X4-UMI. Trees are made from unique V3 genotypes aligned with MUSCLE using Geneious 8.0.3 and generated with DIVEIN PhyML v3.3 (https://indra.mullins.microbiol.washington.edu/DIVEIN/diver.html) [28]. Unique V3 genotypes were often comprised of multiple templates, defined by multiple unique molecular identifiers (UMI). Frequency of each variant as determined by UMI analysis is shown in colored circles according to the color gradient chart, and by the number at the far right of the sequence name. Tropism as determined by geno2pheno FPR cut-off is shown by the text color of each sequence name. Sequences predicted to be X4 by PSSM and codon analysis are indicated in the name of each sequence (“PSSM”, and “C#” where the # corresponds to the codon positions). The last two values in each sequence name, separated by underscores, correspond to the sequence identifier, then the quantity of templates associated with that unique V3 variant as determined by UMI analysis. (a) Specimen #49 was categorized as X4 by Trofile® and R5 by all genotypic analyses (given that plasma HIV RNA load as determined by our qPCR of the specimen we sequenced differed by >1 log10 from that reported to us, and the tropism calls by Trofile® differed from all genotypic interpretation algorithms we suspect a specimen mix-up occurred at some point prior to our specimen processing). (b) Specimen #52 was categorized as X4 by Trofile®, PSSM, codon analysis, and geno2pheno cut-offs ≥3% FPR, but was R5 by geno2pheno 2% FPR. (c) Specimens #45 and #46 were both categorized as R5 by Trofile®; geno2pheno classified both specimens as R5 at FPR 2% but transitioned to X4 by FPR 3%. #45 was classified as R5 and #46 as X4 by PSSM and codon analysis; demonstrating the potential added value of the continuous scale used by Geno2pheno.